# OPTIMIZATION INSIGHTS INTO DEEP DIAGONAL LINEAR NETWORKS

## ABSTRACT

Overparameterized models trained with (stochastic) gradient descent are ubiquitous in modern machine learning. These large models achieve unprecedented performance on test data, but their theoretical understanding is still limited. In this paper, we take a step towards filling this gap by adopting an optimization perspective. More precisely, we study the implicit regularization properties of the gradient flow "algorithm" for estimating the parameters of a deep diagonal neural network. Our main contribution is showing that this gradient flow induces a mirror flow dynamic on the model, meaning that it is biased towards a specific solution of the problem depending on the initialization of the network. Along the way, we prove several properties of the trajectory.

## 1 INTRODUCTION

In recent years, the application of deep networks has revolutionized the field of machine learning, particularly in tasks involving complex data such as images and natural language. These models, typically trained using stochastic gradient descent, have demonstrated remarkable performance on various benchmarks, raising questions about the underlying mechanisms that contribute to their success. Despite their practical efficacy, the theoretical understanding of these models remains relatively limited, creating a pressing need for deeper insights into their generalization abilities. The classical theory shows that the latter is a consequence of regularization, which is the way to impose a priori knowledge into the model and to favour "simple" solutions. While usually regularization is achieved either by choosing simple models or explicitly adding a penalty term to the empirical risk during training, this is not the case for deep neural networks, which are trained simply by minimizing the empirical risk. A new perspective has then emerged in the recent literature, which relates regularization directly to the optimization procedure (gradient based methods). The main idea is to show that the training dynamics themselves exhibit self regularizing properties, by inducing an implicit regularization (bias) which prefers generalizing solutions (see Vardi (2023) for an extended review of the importance of implicit bias in machine learning).

In this context, a common approach is to study simplified models that approximate the networks used in practice. Analyzing the implicit bias of optimization algorithms for such networks is facilitated but still might give some insights on the good performance of neural networks in various scenarios.

### 1.1 RELATED WORKS

**Matrix factorization**    A step in this direction is made by Gunasekar et al. (2017) who analyze the implicit regularization induced by matrix factorization for minimizing a simple least squares objective as $\mathcal{L}(\Theta) = \|A\Theta - b\|^2$ for $\Theta \in \mathbb{R}^{n \times n}$. More precisely, the authors consider the reparametrization $\Theta = UU^T$ for $U \in \mathbb{R}^{n \times d}$ and so the new problem

$$\min_{U \in \mathbb{R}^{n \times d}} \left\| A\left(UU^T\right) - b \right\|^2. \tag{1}$$

Under some assumptions on the initialization, they demonstrate that using gradient flow on the variable $U$ returns a matrix $\Theta = UU^T$ approximating a specific solution of the original problem, the one with minimal nuclear norm. In other words, a simple change of variable before running the algorithm allows to promote sparsity of the recovered solution, which tends to imply a better generalization. Such a setting can roughly be linked to neural networks and has been followed by

several works enriching the understanding of training factorized matrices: Li et al. (2018) study the optimization aspects of such a model; Arora et al. (2018) show that such a reparameterization can improve the conditioning of the problem, accelerating the resulting optimization problem; Gunasekar et al. (2018a) give implicit bias results for an overparameterized matrix factorization; Arora et al. (2019) study the effect of deep matrix factorization, proving a similar implicit bias towards a minimal nuclear norm solution for sufficiently small initialization.

**Weight normalization** Similarly, weight normalization is a strategy used in machine learning that has been studied using this framework. It has been proven to accelerate training (Salimans & Kingma (2016)) and to have desirable implicit regularization properties (Wu et al. (2020)). Moreover, it can be calibrated to take full advantage of both characteristic (Chou et al. (2024)).

**More complex parameterizations** Several works focus on more complex architectures, showing valuable but less interpretable features; for instance, studying the implicit bias in the case of linear convolutional networks (Gunasekar et al. (2018b)), shallow neural networks (Allen-Zhu et al. (2019a;b)), ReLU networks (Vardi & Shamir (2021)) and deep linear networks (Marion & Chizat (2024)). We can also mention the work of Chizat & Bach (2020), where a two-layer neural network with infinite width is considered.

**Diagonal Linear Networks** Diagonal Linear Networks are introduced in the literature by Woodworth et al. (2020) and Moroshko et al. (2020) as a linear approximation of a neural network with diagonal connections between the nodes (see Figure 1). In this case the network function is given by $f_\theta(x) = \theta^T x$ with the parameters vector expressed as $\theta = u \odot v$, where $\odot$ denotes the Hadamard product. Although this model appears to be overly simplistic, its analysis offers a considerable degree of insight. Both Woodworth et al. (2020) and Moroshko et al. (2020) highlight the role of the initialization for such a network in regard of the implicit bias induced on the solution: a small initialization favours a solution close to that with minimal $L^1$ norm, while a large one tends to minimize the $L^2$ norm. These two regimes can be put in parallel with respectively the rich regime and the kernel (or lazy) regime outlined in Chizat et al. (2019). Azulay et al. (2021) give additional results on the implicit bias of the network, allowing untied weights and also considering the case of connected networks (non diagonal). This model has been also been analyzed to better understand the implicit bias induced by stochastic gradient descent Pesme et al. (2021); Even et al. (2023), by the choice of the step size Nacson et al. (2022) and by other methods involving momentum Papazov et al. (2024). The $L$-layer case has been studied by Woodworth et al. (2020); Moroshko et al. (2020); Chou et al. (2023a) for a very simplified model, suggesting that the number of layers is positively correlated with the sparsity of the selected solution.

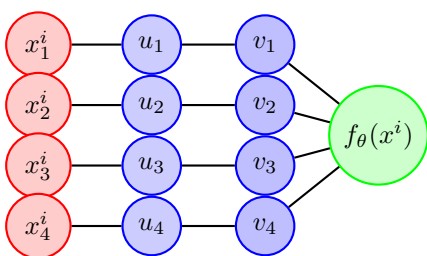

Figure 1: Representation of a Diagonal Linear Network

**Hadamard parameterization to promote sparsity** Diagonal Linear Networks are closely related to Hadamard parameterization (HP) and it is worth noticing that HP was used before for sparsity recovery. Hoff (2017) takes benefit of HP to promote sparsity in a LASSO problem. Vaskevicius et al. (2019); Zhao et al. (2022) also consider an HP for a least squares problem inducing an implicit bias towards the minimal $L^1$-norm solution, enabling the use of early stopping strategies. In a comparable vein, Amid & Warmuth (2020) exploit a reparameterization $\theta = u^{\odot 2}$ for a classification problem and Chou et al. (2022) apply HP to solve non negative least squares. Poon & Peyré (2023) elegantly reformulate a group-LASSO problem using some Hadamard parameterization in order to

apply more efficient optimization methods. The geometry induced by HP is also a subject of interest and in this regard, Ouyang et al. (2024) give necessary condition for a reparameterized function to satisfy the Kurdyka-Łojasiewicz property.

**Links with Mirror Flow**    As shown in Azulay et al. (2021); Chou et al. (2023a), Diagonal Linear Networks trained with a Gradient Flow (or Gradient Descent) share direct links with Mirror Flow (or Mirror Descent). One of the first appearances of the Mirror Flow dynamic can be attributed to Alvarez et al. (2004). Many years later, a number of studies are presented in the context of understanding the effects of reparameterization. The mirror gradient method having an implicit bias towards specific solutions, Vaskevicius et al. (2020) propose early stopping strategies taking advantage of it. Li et al. (2022) provide assumptions that are necessary for a reparameterization trained with a Gradient Flow to be equivalent to a Mirror Flow. In their paper, Chou et al. (2023b) adopt an original point of view, seeing reparameterization as a way to enforce some implicit bias. In this perspective, the authors give guidelines to define it efficiently.

## 1.2 CONTRIBUTIONS

The general problem stated at the beginning of the introduction can be phrased in a precise mathematical form. We want to study the properties of an optimization procedure run on the following problem, that is the problem available in practice (e.g. an empirical risk depending on some parameters):

$$\min_{u \in \mathbb{R}^p} \tilde{\mathcal{L}}(u). \tag{2}$$

We look at this through the lens of reparameterization: we suppose that the previous problem arises from a hidden problem through some change of variable, i.e. that there exists a loss function $\mathcal{L} : \mathbb{R}^d \to \mathbb{R}$ and a reparametrization $q : \mathbb{R}^p \to \mathbb{R}^d$ such that

$$\tilde{\mathcal{L}}(u) = \mathcal{L}(q(u)).$$

In the variable $\theta = q(u)$, the hidden problem is then given by

$$\min_{\theta \in \mathbb{R}^d} \mathcal{L}(\theta). \tag{3}$$

The general question of this line of research is the following. Running an optimization procedure on the reparametrization $u \in \mathbb{R}^p$ to minimize the loss $\tilde{\mathcal{L}}$ of the available problem in 2, what happens on the variable $\theta = q(u)$ in terms of the minimization of $\mathcal{L}$, the hidden problem in 3?
One example of the previous setting is given by Shallow Linear (connected) Neural Networks, where, for $u = (W, v)$, $q(W, v) = W^T v$.

In this paper we focus on a Deep Diagonal Linear Networks, an extension of Diagonal Linear Networks to the $L$-layer case. The corresponding reparameterization of $\theta \in \mathbb{R}^d$ is given by

$$\theta = q(u^1, u^2, \dots, u^L) = u^1 \odot u^2 \odot \cdots \odot u^L,$$

where $L$ is the number of layers and $u^i \in \mathbb{R}^d$ for every $i = 1, ..., L$. In this case, considering for instance the empirical risk with linear-quadratic loss on input-output data $(x_i, y_i)_{i=1}^n$, we have

$$\min_{(u^1, u^2, \dots, u^L)} \tilde{\mathcal{L}}(u^1, u^2, \dots, u^L) = \mathcal{L}(q(u^1, u^2, \dots, u^L)) = \sum_{i=1}^n (\langle u^1 \odot u^2 \odot \cdots \odot u^L \rangle, x_i \rangle - y_i)^2,$$

where $\mathcal{L}(\theta) = \sum_{i=1}^n (\langle \theta, x_i \rangle - y_i)^2$.

Although simplified versions of this model have been previously studied in the literature (see Woodworth et al. (2020); Moroshko et al. (2020); Chou et al. (2023a)), this formulation appears to better approximate real-life networks. In fact, these works consider the reparameterization $\theta = u^{\odot L} - v^{\odot L}$, allowing for the derivation of numerous properties. Nevertheless, it can be argued that this model is overly simplistic, as the number of layers does not affect the number of hyperparameters.

The objective of this work is to gain a deeper comprehension of the effect of reparameterization when the network layers $(u^j)_{j \in [L]}$ are trained using Gradient Flow. In this regard, we make the following contributions:

- We identify quantities that are conserved along the trajectory of the flow. This allows us to show that the dynamic of the variable $\theta$ is driven by that of an abstract layer containing the minimal nodes.

- We provide a mild assumption on the initialization such that applying Gradient Flow to the layers $(u^j)_{j \in [L]}$ generates a trajectory in the variable $\theta$ that is the solution of a Mirror Flow.

- We show several convergence properties in terms of the original optimization problem, under the same assumption. In particular, we show that the value of the loss function along the variable $\theta$ decreases exponentially to its minimal value under a Polyak-Łojasiewicz property. These results also reveal that initializing the layers with small values leads to a slow training, a phenomenon that is known in the literature for other models.

### 1.3 NOTATION

For any positive integer $L$, the set of integers from 1 to $L$ is denoted by $[L]$. The coordinate-wise product, also called the Hadamard product, of two vectors $x$ and $y$ in $\mathbb{R}^p$ is denoted $x \odot y$, where for each $i$ in $[p]$, $(x \odot y)_i = x_i y_i$. The vector $x^{\odot L}$ is defined as $\left(x_i^L\right)_{i \in [p]}$ and if $\left(x^j\right)_{j \in [L]} \in (\mathbb{R}^p)^L$, then $\bigodot_{j=1}^L x^j$ refers to $\left(\prod_{j=1}^L x_i^j\right)_{i \in [p]}$. We use $|\cdot|$ to denote the absolute value which can be defined for vectors in $\mathbb{R}^p$: for any $x \in \mathbb{R}^p$, $|x| = (|x_i|)_{i \in [p]}$. Similarly, the sign function denoted $\text{sign}(\cdot)$, the square root function $\sqrt{\cdot}$, the hyperbolic sine $\sinh$, the inverse hyperbolic sine $\text{arcsinh}$ and the logarithm $\log$ are considered as functions from $\mathbb{R}^p$ to $\mathbb{R}^p$ that apply to each component. The Jacobian of $G = (G_1, \ldots, G_p)$ with respect to $\theta \in \mathbb{R}^k$ is denoted by $J_G(\theta)$ and is defined as:

$$J_G(\theta) := \left[\frac{\partial G}{\partial \theta_1}; \ldots; \frac{\partial G}{\partial \theta_k}\right] = \begin{bmatrix} \frac{\partial G_1}{\partial \theta_1} & \cdots & \frac{\partial G_1}{\partial \theta_k} \\ \vdots & \ddots & \vdots \\ \frac{\partial G_p}{\partial \theta_1} & \cdots & \frac{\partial G_p}{\partial \theta_k} \end{bmatrix}.$$

The Hessian of $G$ is denoted by $\nabla^2 G$. We denote by $\#$ the cardinality of a set, and by $\%$ the modulo operation. The vector $\mathbf{1} \in \mathbb{R}^p$ denotes the unitary vector equal to $(1)_{i \in [p]}$ and $\lfloor \cdot \rfloor$ represents the floor function. For a vector $v \in \mathbb{R}^p$, $\text{diag}(v)$ denotes the diagonal matrix in $\mathbb{R}^{p \times p}$ having the diagonal equal to $v$.

## 2 DEEP DIAGONAL LINEAR NETWORKS

In this section, we state several results on the trajectory of the variable $\theta$ when parameterized by a Deep Diagonal Linear Network and when its parameters are trained by Gradient Flow. More specifically, we overparameterize our system by defining

$$\theta = \bigodot_{j=1}^L u^j,$$

where $L \geqslant 2$ is the number of layers of the network and each $u^j \in \mathbb{R}^d$ corresponds to the $j$-th layer, as illustrated in Figure 2. Note that for $L = 2$, we recover the classical Diagonal Linear Network. We consider the training of the Deep Diagonal Linear Network by Gradient Flow, which can be written as follows:

$$\forall j \in [L], \ \forall t \geqslant 0, \quad \frac{du^j(t)}{dt} + \nabla_j \mathcal{L}\left(\bigodot_{k=1}^L u^k(t)\right) = 0, \tag{4}$$

where we denoted $\left(\nabla_j \mathcal{L}\left(\bigodot_{k=1}^L u^k\right)\right)_i = \frac{\partial}{\partial u_i^j} \mathcal{L}\left(\bigodot_{k=1}^L u^k\right)$ for any $j \in [L]$ and $i \in [d]$. By simple computations, we get the following dynamical system:

$$\forall j \in [L], \ \forall t \geqslant 0, \quad \frac{du^j(t)}{dt} + \left(\bigodot_{k \neq j} u^k(t)\right) \odot \nabla \mathcal{L}(\theta(t)) = 0. \tag{5}$$

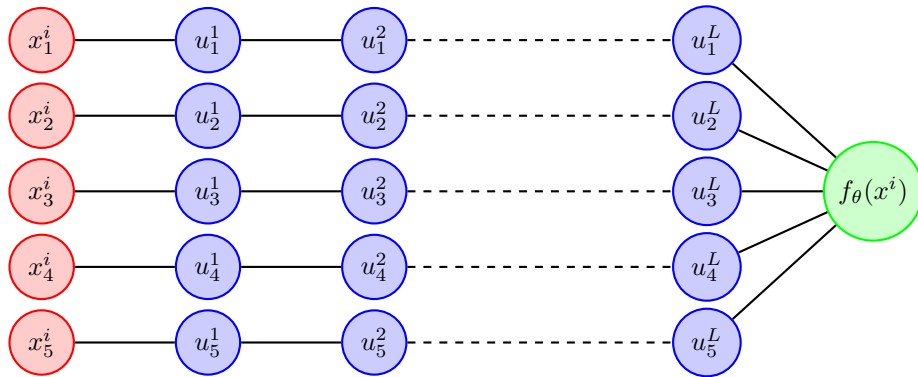

Figure 2: Representation of a Deep Diagonal Linear Network with $L$ layers in $\mathbb{R}^d$, $d = 6$.

We first derive several properties of the trajectories of the weights of the network. We then demonstrate the main result on the dynamic of $\theta$ by adopting a different approach. Convergence guarantees are stated in the last section, under the same assumption on the initialization.

## 2.1 ON THE BEHAVIOR OF THE FLOW

From equation 5, we can prove the following properties on the trajectory of the layers.

**Proposition 1.** *Let $\left(u^j\right)_{j \in [L]}$ satisfy equation 5 and let $\theta = \bigodot_{j=1}^{L} u^j$. Then the following statements hold:*

*1. For any $(j, k) \in [L] \times [L]$,*

$$\forall t \geqslant 0, \ \frac{d}{dt}\left(u^j \odot u^j\right)(t) = \frac{d}{dt}\left(u^k \odot u^k\right)(t) = -\theta(t) \odot \nabla L(\theta(t)). \tag{6}$$

*In particular,*

$$\forall t \geqslant 0, \ u^j(t)^{\odot 2} - u^j(0)^{\odot 2} = u^k(t)^{\odot 2} - u^k(0)^{\odot 2}. \tag{7}$$

*2. Let $i \in [d]$. If $j \notin \arg\min_{k \in [L]} |u_i^k(0)|$, then for every $t > 0$ we have that $u_i^j(t) \neq 0$.*

**Proof.**
1. This statement is obtained by multiplying equation 5 by $u^j(t)$ and then by integrating it.
2. Let $i \in [d]$ and $j \notin \arg\min_{k \in [L]} |u_i^k(0)|$. We suppose that there exists $T > 0$ such that $u_i^j(T) = 0$. We define $l \in \arg\min_{k \in [L]} |u_i^k(0)|$ and according to the first claim, for all $t \geqslant 0$,

$$u_i^l(t)^2 - u_i^l(0)^2 = u_i^j(t)^2 - u_i^j(0)^2.$$

In particular, for $t = T$,

$$u_i^l(T)^2 = u_i^l(0)^2 + u_i^j(T)^2 - u_i^j(0)^2 = u_i^l(0)^2 - u_i^j(0)^2 < 0.$$

Since the above equation cannot be true, we can conclude. $\qquad \square$

Before commenting the previous result, we introduce the main assumption on the initialization of the network that we will consider in the next:

$$\forall i \in [d], \ \#\left(\arg\min_{k \in [L]} |u_i^k(0)|\right) = 1. \tag{$\mathcal{A}$}$$

The preceding condition indicates that, for each component $i \in [d]$, the node with the minimal absolute value is unique across all layers. For instance, initializing each layer with the same values violates this assumption; while setting them randomly is almost surely valid.

It should be noted that this initialization assumption is relatively weak in comparison to those presented in the existing literature, even when considering only two layers. For Diagonal Linear Networks, Woodworth et al. (2020) impose a layer to be zero before training while in the context of deep matrix factorization, Arora et al. (2019) impose the layers to be balanced i.e.

$$\forall (i,j) \in [L]^2, \quad W_i(0)^T W_i(0) = W_j(0)^T W_j(0),$$

where $W_j$ is the matrix containing the weights of the $j$-th layer.

One can notice that by assuming $\mathcal{A}$, the second claim of Proposition 1 ensures that for each component $i$, only one layer $j$ can cross zero (and this layer is the one with the minimal absolute value at initialization). This phenomenon can be observed in Figure 3 where for the first component, only the third layer changes its sign throughout the training process. This layer is that with minimal absolute value at initialization.

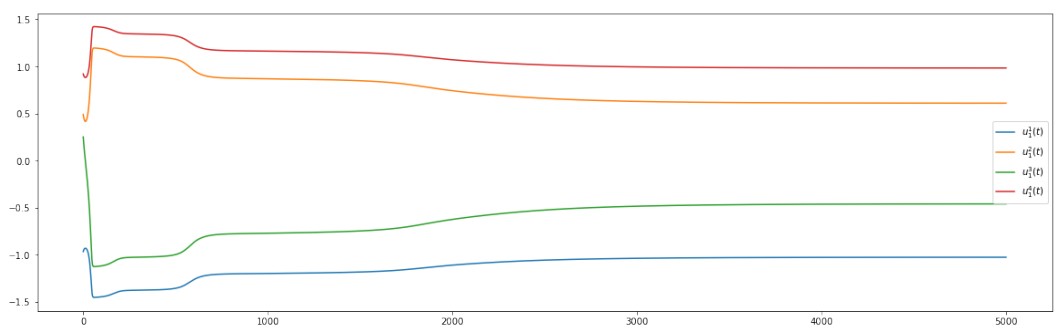

Figure 3: Dynamic of the nodes $(u_1^j(t))_{j \in [L]}$ for a loss function satisfying $\mathcal{L} : \theta \mapsto \|X\theta - y\|^2$ with $X \in \mathbb{R}^{10 \times 5}$ and $y \in \mathbb{R}^{10}$ generated randomly, and $L = 4$ layers.

We use this feature to rewrite the dynamic of the layers.

**Simplifying the system** Suppose that assumption $\mathcal{A}$ is satisfied. Then, it allows us to define $(v^j)_{j \in [L]} \in (\mathbb{R}^d)^L$ as a permutation of $(u^j)_{j \in [L]}$ in the following way: for any $i \in [d]$,

$$v_i^1 = u_i^j \text{ where } j = \arg \min_{k \in [L]} |u_i^j(0)|,$$

and for any $j \in [\![2, L]\!]$,

$$v_i^j = \begin{cases} u_i^1 \text{ if } j = \arg \min_{k \in [L]} |u_i^j(0)|. \\ u_i^j \text{ else.} \end{cases}$$

In other words, $(v^j)_{j \in [L]}$ is a permutation of $(u^j)_{j \in [L]}$ where the first layer gathers all minimal absolute values at initialization. Note that we have $\theta = \bigodot_{j=1}^{L} v^j$ and it is trivial to show that the equations and properties stated above still hold for $(v^j)_{j \in [L]}$.

Since we assume that $\mathcal{A}$ holds, the second claim of Proposition 1 ensures that for any $j \in [\![2, L]\!]$,

$$\forall t \geqslant 0, \ \text{sign}\left(v^j(t)\right) = \text{sign}\left(v^j(0)\right),$$

where sign $: \mathbb{R}^d \to \mathbb{R}^d$ returns the sign for every component. By using the first claim of Proposition 1, we get that

$$\forall j \in [\![2, L]\!], \ \forall t \geqslant 0, \ v^j(t) = \text{sign}\left(v^j(0)\right) \odot \sqrt{v^1(t)^{\odot 2} + v^j(0)^{\odot 2} - v^1(0)^{\odot 2}}.$$

By rewriting equation 5 for the first layer, we have that:

$$\forall t \geqslant 0, \ \frac{dv^1(t)}{dt} + \text{sign}\left(\bigodot_{j \neq 1} v^j(0)\right) \odot \left(\bigodot_{j \neq 1} \sqrt{v^1(t)^{\odot 2} + \Delta_j}\right) \odot \nabla L(\theta(t)) = 0, \quad (8)$$

where $\Delta_j = v^j(0)^{\odot 2} - v^1(0)^{\odot 2}$ for any $j \in [d]$. Since

$$\theta(t) = \text{sign}\left(\bigodot_{j=1}^{L} v^j(0)\right) \odot \left(\bigodot_{j=1}^{L} \sqrt{v^1(t)^{\odot 2} + \Delta_j}\right),$$

we can see that **the flow of $\theta$ is entirely determined by that of $v^1$ and the initialization of the network**.

**Deducing a Mirror Flow?** From that point, it appears natural to apply the strategy proposed by Woodworth et al. (2020) to study Diagonal Linear Networks: the authors leverage a formulation of $\theta$ to derive an implicit bias which is also linked to a Mirror Flow (for further details, see Appendix A.1; for another example to which it applies, see Appendix A.3). However, if $L > 2$, equation 8 does not allow us to get an analytical expression of $v^1(t)$ and thus of $\theta(t)$. As a consequence, this approach cannot be applied in this situation. Note that in the case $L = 2$, equation 8 gives for any $t \geqslant 0$,

$$\frac{dv^1(t)}{dt} + \text{sign}\left(v^2(0)\right) \odot \sqrt{v^1(t)^{\odot 2} + \Delta} \odot \nabla L(\theta(t)) = 0, \tag{9}$$

with $\Delta = v^2(0)^{\odot 2} - v^1(0)^{\odot 2}$, allowing to compute $\theta$ and recover the analysis of Woodworth et al. (2020).

## 2.2 IT IS ACTUALLY A MIRROR FLOW

Since this setting does not fit the framework of Woodworth et al. (2020), we base our analysis on the work of Li et al. (2022) to prove the following theorem. Although our analysis demonstrates that $\theta$ is the solution of a Mirror Flow, it does not yield the associated entropy function (nor the implicit bias).

**Theorem 1.** *Let $\theta \in \mathbb{R}^d$ be parameterized as $\theta = \bigodot_{j=1}^{L} u^j$ and $\left(u^j\right)_{j \in [L]}$ follow the dynamic described in equation 5. If the initialization of the network satisfies $\mathcal{A}$, then $\theta$ follows a Mirror Flow dynamic, i.e there exists a convex entropy function $\mathcal{Q} : \mathbb{R}^d \to \mathbb{R}$ such that:*

$$\forall t \geqslant 0, \quad \frac{d\nabla \mathcal{Q}(\theta(t))}{dt} + \nabla \mathcal{L}(\theta(t)) = 0.$$

**Proof.**
The proof of this claim consists in showing that the reparameterization induced by Deep Diagonal Linear Networks satisfies the hypotheses necessary to the application of (Li et al., 2022, Theorem 4.8). It is first necessary to introduce some notation.

Let $\theta \in \mathbb{R}^d$ be parameterized as $\theta = \bigodot_{j=1}^{L} u^j$ where $\left(u^j\right)_{j \in [L]}$. We denote $\boldsymbol{u} \in \mathbb{R}^{L \times d}$ the entire set of parameters of the network. More specifically, we have

$$\forall n \in [Ld], \ \boldsymbol{u}_i = u_{\lfloor (i-1)/L \rfloor + 1}^{(i-1)\%L+1},$$

meaning that we can write

$$\boldsymbol{u} = \left(u_1^1 \ u_1^2 \ \ldots \ u_1^L \ u_2^1 \ \ldots\ldots\ldots \ u_d^1 \ \ldots \ u_d^L\right)^T.$$

It is then possible to write $\theta$ according to $\boldsymbol{u}$ via the parameterization function $G : \mathbb{R}^{L \times d} \to \mathbb{R}^d$ defined as follows

$$\forall w \in \mathbb{R}^{L \times d}, \ \forall i \in [d], \ G_i(w) = \prod_{j=(i-1)L+1}^{iL} w_j. \tag{10}$$

Showing the desired claim then requires to prove that $G : \mathcal{M} \to \mathbb{R}^d$ is a commuting and regular parameterization for a well-chosen smooth manifold $\mathcal{M}$ of $\mathbb{R}^{L \times d}$. Then, under an additional technical assumption that we discuss and if $\boldsymbol{u}(0) \in \mathcal{M}$, (Li et al., 2022, Theorem 4.8) ensures that the trajectory of $\theta$ is the solution of a Mirror Flow. The detailed computations are given in Appendix A.4. $\square$

In the previous theorem we have shown that, under mild assumptions on the inizialization,

running Gradient Flow on the reparametrization $(u^j)_{j \in [L]}$ given by a Deep Diagonal Linear Network generates a trajectory in $\theta(t) = \odot_{j=1}^L u^j(t)$ that is the solution of a Mirror Flow on the function $\mathcal{L}$ with entropy $\mathcal{Q}$. The strategy adopted does not lead to an explicit expression for the entropy $\mathcal{Q}$ (and so for the implicit bias induced by the reparametrization). On the other hand, we can guarantee that the entropy $\mathcal{Q}$ is convex and so, for instance, that the quantity $\mathcal{L}(\theta(t))$ is non-increasing along time.

## 2.3 Convergence guarantees

We now give additional convergence results on the trajectory of $\theta$ under the assumption $\mathcal{A}$. First, notice that $\theta$ is a solution of the following dynamical system

$$\forall t \geqslant 0, \quad \dot{\theta}(t) + \sum_{j=1}^L \left( \bigodot_{k \neq j} u^k(t)^{\odot 2} \right) \odot \nabla \mathcal{L}(\theta(t)) = 0. \tag{11}$$

We can easily write this equation in the following way:

$$\forall t \geqslant 0, \quad \dot{\theta}(t) + M(t) \nabla \mathcal{L}(\theta(t)) = 0, \tag{12}$$

where $M : t \mapsto \text{diag}\left( \left( \sum_{j=1}^L \left( \prod_{k \neq j} u_i^k(t)^2 \right) \right)_{i \in [d]} \right)$.

Since we assume that assumption $\mathcal{A}$ holds, the second claim of Proposition 1 ensures that for any $i \in [d]$, if $j \notin \arg\min_{k \in [d]} |u_i^k(0)|$, then $|u_i^j(t)| > 0$ for all $t \geqslant 0$. More precisely, using the first claim of Proposition 1, we obtain that $u_i^j(t)^2 \geqslant u_i^j(0)^2 - u_i^l(0)^2$ where $l = \arg\min_{k \in [d]} |u_i^k(0)|$. Therefore, for any $i \in [d]$,

$$\forall t \geqslant 0, \sum_{j=1}^L \left( \prod_{k \neq j} u_i^k(t)^2 \right) = \sum_{j=1, \, j \neq l}^L \left( \prod_{k \neq j} u_i^k(t)^2 \right) + \prod_{k \neq l} u_i^k(t)^2$$
$$\geqslant 0 + \prod_{k \neq l} (u_i^k(0)^2 - u_i^l(0)^2) > 0. \tag{13}$$

As a consequence, we get the following properties:

- $M(t)$ is invertible for all $t \geqslant 0$ and

$$M^{-1}(t) = \text{diag}\left( \left( \left( \sum_{j=1}^L \left( \prod_{k \neq j} u_i^k(t)^2 \right) \right)^{-1} \right)_{i \in [d]} \right).$$

  As a consequence,

$$\forall t \geqslant 0, \quad M^{-1}(t) \dot{\theta}(t) + \nabla \mathcal{L}(t) = 0,$$

  which means that the mirror map $\mathcal{Q}$ from Theorem 1 satisfies for any $t \geqslant 0$

$$\frac{d \nabla \mathcal{Q}(\theta(t))}{dt} = M^{-1}(t) \dot{\theta}(t).$$

- The smallest eigenvalue $\lambda_{min}(M(t))$ of $M(t)$ admits a lower bound that does not depend on $t$:

$$\lambda_{min}(M(t)) \geqslant \min_{i \in [d]} \prod_{k \neq k_i} (u_i^k(0)^2 - u_i^{k_i}(0)^2), \tag{14}$$

  where $k_i = \arg\min_{k \in [d]} |u_i^k(0)|$ for any $i \in [d]$.

Thus, we get the following result which applies, for example, if $\mathcal{L}$ is the squared loss function.

**Theorem 2.** *Let $\theta \in \mathbb{R}^d$ be parameterized as $\theta = \bigodot_{j=1}^{L} u^j$ and $\left(u^j\right)_{j \in [L]}$ follow the dynamic described in equation 5. Suppose that $\arg\min_{\theta \in \mathbb{R}^d} \mathcal{L}(\theta) \neq \emptyset$ and let $\mathcal{L}^*$ denote $\min_{\theta \in \mathbb{R}^d} \mathcal{L}(\theta)$. If the initialization of the network satisfies $\mathcal{A}$ and the loss function $\mathcal{L}$ satisfies a Polyak-Łojasiewicz property with parameter $\mu > 0$, then for all $t \geqslant 0$,*

$$\mathcal{L}(\theta(t)) - \mathcal{L}^* \leqslant e^{-2\sigma\mu t} \left(\mathcal{L}(\theta(0)) - \mathcal{L}^*\right),$$

*where $\sigma = \min_{i \in [d]} \prod_{k \neq k_i} (u_i^k(0)^2 - u_i^{k_i}(0)^2)$ and $k_i = \arg\min_{k \in [d]} |u_i^k(0)|$ for any $i \in [d]$.*

**Proof.**
For all $t \geqslant 0$, we have that:

$$\begin{aligned}
\frac{d}{dt}\left(\mathcal{L}(\theta(t)) - \mathcal{L}^*\right) &= \left\langle \nabla\mathcal{L}(\theta(t)), \dot{\theta}(t) \right\rangle \\
&= -\left\langle \nabla\mathcal{L}(\theta(t)), M(t)\nabla\mathcal{L}(\theta(t)) \right\rangle \\
&\leqslant -\lambda_{min}(M(t))\|\nabla\mathcal{L}(\theta(t))\|^2 \\
&\leqslant -\sigma\|\nabla\mathcal{L}(\theta(t))\|^2,
\end{aligned}$$

where we use the lower bound $\sigma = \min_{i \in [d]} \prod_{k \neq k_i} (u_i^k(0)^2 - u_i^{k_i}(0)^2)$. is define in equation 14. Supposing that $\mathcal{L}$ satisfies the Polyak-Łojasiewicz inequality, there exists $\mu > 0$ such that:

$$\forall \theta \in \mathbb{R}^d, \quad 2\mu\left(\mathcal{L}(\theta) - \mathcal{L}^*\right) \leqslant \|\nabla\mathcal{L}(\theta)\|^2.$$

In particular, this guarantees that:

$$\frac{d}{dt}\left(\mathcal{L}(\theta(t)) - \mathcal{L}^*\right) \leqslant -2\sigma\mu\left(\mathcal{L}(\theta(t)) - \mathcal{L}^*\right),$$

and as a consequence:

$$\forall t \geqslant 0, \quad \mathcal{L}(\theta(t)) - \mathcal{L}^* \leqslant e^{-2\sigma\mu t}\left(\mathcal{L}(\theta(0)) - \mathcal{L}^*\right). \tag{15}$$

$\square$

This bound on the suboptimality gap reveals that the initialization of the network significantly affects the convergence rate of its training. In this regard, it is erroneous to assume that the linear rate necessarily implies rapid convergence, as the parameter $\sigma = \min_{i \in [d]} \prod_{k \neq k_i} (u_i^k(0)^2 - u_i^{k_i}(0)^2)$ plays an important role. If the gap between the minimal absolute value of a component across layers and the others is small, then $\sigma$ is also considerably small, which affects the speed of convergence during training. Figure 4 emphasizes this point, showing that larger values at initialization lead to a better convergence rate.

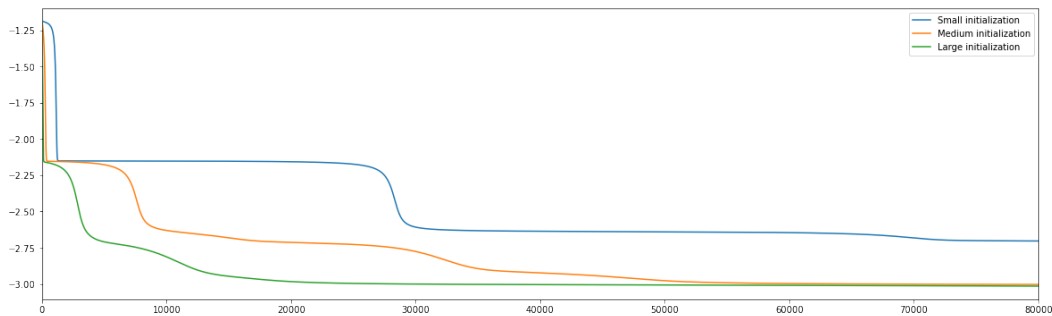

Figure 4: Evolution of $\log\left(\mathcal{L}(\theta(t)) - \mathcal{L}^*\right)$ according to time for three 6-layer networks with different initialization. The loss function is defined as $\mathcal{L} : \theta \mapsto \|X\theta - y\|^2$ with $X \in \mathbb{R}^{10 \times 8}$ and $y \in \mathbb{R}^{10}$ generated randomly. Each network is initialized with a first layer having components equal to $0$. The initial value of the remaining layers of the first network (in blue) is generated randomly, while that of the second (in orange) and the third (in green) are respectively equal to $1.4$ and $1.8$ times the values of the firs component-wise.

This phenomenon has been documented in the literature for similar models (e.g. see Chou et al. (2023a; 2024)) and may indicate that initializing the network with larger values would be beneficial.

However, it is also well known that the initialization of a network should be chosen small enough to induce sparsity of the approximated solutions and consequently generalize well, see Gunasekar et al. (2017); Arora et al. (2019); Woodworth et al. (2020); Moroshko et al. (2020); Chizat et al. (2019). To address the challenge of slow training, Chou et al. (2024) suggest employing weight normalization, a reparameterization technique that promotes comparable sparsity in the solution while ensuring reasonable convergence rates.

## 3 CONCLUSION AND PERSPECTIVES

In this work, we have studied Deep Diagonal Linear Networks and broaden our understanding of their training. In particular, we have analyzed the trajectories of the original variable when a Gradient Flow is applied to its layers. In this way, we were able to extract a sufficient initialization condition to prove theoretical results both on the optimization and the bias of this model. We show indeed that such a training induces a Mirror Flow on the original variable and ensures a linear decrease of the error on the loss function as long as it satisfies the Polyak-Łojasiewicz inequality. This study also reveals the negative impact of initializing the network with small values in terms of convergence speed.

Despite its apparent simplicity, this model is still tricky to study and we were unable to compute the entropy function associated to the Mirror Flow induced. It is then a natural question remaining that would also inform us on the implicit bias underlying. It is also noteworthy that the linearity of this model may prevent us from fully understanding the generalization properties of real-world neural networks. This highlights the potential necessity for incorporating non-linearity in future works. From an optimization perspective, further research could be conducted to investigate the influence of stochasticity, momentum, and more generally, the optimization method applied during training.

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

# A APPENDIX

## A.1 BACKGROUND: A DIRECT STRATEGY TO ANALYZE DIAGONAL LINEAR NETWORKS

In this section, we briefly state the strategy proposed by Woodworth et al. (2020) to analyze the training of a Diagonal Linear Network through Gradient Flow. The proposed approach to derive an implicit bias benefits from its simplicity and is general enough to be extended to other simple models. To emphasize this point, we apply it also to Deep Redundant Linear Network in Appendix A.3.

We are interested in solving the problem:

$$\min_{\theta \in \mathbb{R}^d} \mathcal{L}(\theta),$$

for some loss function $\mathcal{L} : \mathbb{R}^d \to \mathbb{R}$, where the loss is minimal if $X\theta = y$, $X : \mathbb{R}^d \to \mathbb{R}^n$ and $y \in \mathbb{R}^n$ denoting some data input and output. We look at this problem through the lens of reparameterization, i.e. we consider the change of variable $\theta = u \odot v$ which corresponds to a Diagonal Linear Network. Such a network is then trained using a Gradient Flow on the new parameters:

$$\begin{cases} \dot{u}(t) + v(t) \odot \nabla \mathcal{L}(\theta(t)) = 0 \\ \dot{v}(t) + u(t) \odot \nabla \mathcal{L}(\theta(t)) = 0. \end{cases} \tag{16}$$

Further computations, detailed in Appendix A.2, show that the variable $\theta$ can be written in the following way:

$$\theta(t) = \Psi(\xi(t)) := \frac{\left| u(0)^{\odot 2} - v(0)^{\odot 2} \right|}{2} \sinh\left( 2\xi(t) + \log\left| \frac{u(0) + v(0)}{u(0) - v(0)} \right| \right), \tag{17}$$

where $\xi : t \mapsto -\int_0^t \nabla \mathcal{L}(\theta(s))ds$ is the solution of

$$\dot{\xi}(t) + \nabla \mathcal{L}(\theta(t)) = 0.$$

**Deriving the implicit bias** Suppose now that $\lim_{t \to +\infty} \theta(t) = \theta_\infty$ such that $X\theta_\infty = y$, and that the loss function $\mathcal{L}$ can be written as $\mathcal{L}(\theta) = \ell(X\theta - y)$ for any $\theta \in \mathbb{R}^d$ with $\ell : \mathbb{R}^n \to \mathbb{R}$. Proving that the trajectories generated by equation 23 are implicitly biased towards a potential $\mathcal{Q}$ consists in showing that:

$$\theta_\infty = \arg \min_{X\theta = y} \mathcal{Q}(\theta). \tag{18}$$

In order to achieve this, Woodworth et al. (2020) notice that $\theta_\infty = \Psi(X^T \nu)$ (where $\Psi$ is defined in equation 17) for some $\nu \in \mathbb{R}^n$. Indeed,

$$\lim_{t \to +\infty} \xi(t) = -\int_0^{+\infty} \nabla \mathcal{L}(\theta(s))ds = -X^T \underbrace{\int_0^{+\infty} \nabla \ell(X\theta(s) - y)ds}_{:= -\nu}.$$

This property is crucial since by writing the KKT conditions associated to the problem described in equation 18, we obtain:

$$\begin{cases} X\theta_\infty = y \\ \exists \omega \in \mathbb{R}^n, \ \nabla \mathcal{Q}(\theta_\infty) = X^T \omega \end{cases} \tag{19}$$

The first condition is satisfied by assumption. Defining $\mathcal{Q}$ such that $\nabla \mathcal{Q}(\theta) = \Psi^{-1}(\theta)$ for any $\theta$, we get that $\nabla \mathcal{Q}(\theta_\infty) = \Psi^{-1}(\theta_\infty) = X^T \nu$, demonstrating that equation 18 holds. In this case, we have

$$\mathcal{Q}(\theta) = \frac{1}{2} \sum_{i=1}^d \left( 2\theta_i \operatorname{arcsinh} \left( \frac{2\theta_i}{\Delta_0} \right) - \sqrt{4\theta_i^2 + \Delta_0^2} + \Delta_0 \right) - \frac{1}{2} \left\langle \log \left| \frac{u(0) + v(0)}{u(0) - v(0)} \right|, \theta \right\rangle, \tag{20}$$

where $\Delta_0 = \left| u(0)^{\odot 2} - v(0)^{\odot 2} \right|$.

**Revealing a Mirror Flow** The above strategy provides insights on the implicit bias induced by reparameterizing the problem and also allows one to reveal a more general property on the trajectory of $\theta$.

The function $\mathcal{Q}$ defined in equation 20 ensures that for any $\theta$, $\nabla \mathcal{Q}(\theta) = \Psi^{-1}(\theta)$. In particular, for any $t \geqslant 0$:

$$\nabla \mathcal{Q}(\theta(t)) = \Psi^{-1}(\theta(t)) = \xi(t), \tag{21}$$

according to equation 17. As mentionned before, $\dot{\xi}(t) + \nabla \mathcal{L}(\theta(t)) = 0$ and thus:

$$\frac{d\nabla \mathcal{Q}(\theta(t))}{dt} + \nabla \mathcal{L}(\theta(t)) = 0. \tag{22}$$

This shows that $\theta$ is the solution of a Mirror Flow dynamic with mirror map $\mathcal{Q}$.

### A.2 DETAILED COMPUTATIONS FOR DIAGONAL LINEAR NETWORKS

Recall that we focus on the problem:

$$\min_{\theta \in \mathbb{R}^d} \mathcal{L}(\theta),$$

for some loss function $\mathcal{L} : \mathbb{R}^d \to \mathbb{R}$, and we consider the reparameterization $\theta = u \odot v$. By applying a Gradient Flow to the hyperparameters $u$ and $v$ we obtain:

$$\begin{cases} \dot{u}(t) + v(t) \odot \nabla \mathcal{L}(\theta(t)) = 0 \\ \dot{v}(t) + u(t) \odot \nabla \mathcal{L}(\theta(t)) = 0. \end{cases} \tag{23}$$

It is then trivial to observe that $(z_+, z_-)$ defined as $(u + v, u - v)$ is a solution of:

$$\begin{cases} \dot{z}_+(t) + z_+(t) \odot \nabla \mathcal{L}(\theta(t)) = 0 \\ \dot{z}_-(t) - z_-(t) \odot \nabla \mathcal{L}(\theta(t)) = 0 \end{cases}, \tag{24}$$

which guarantees that for any $t \geqslant 0$,

$$\begin{cases} z_+(t) = z_+(0)e^{\xi(t)} \\ z_-(t) = z_-(0)e^{-\xi(t)} \end{cases}, \tag{25}$$

where $\xi : t \mapsto -\int_0^t \nabla\mathcal{L}(\theta(s))ds$. Note by defining $\xi$ in this way, it holds that

$$\dot{\xi}(t) + \nabla\mathcal{L}(\theta(t))) = 0, \quad \xi(0) = 0.$$

It is done by observing that $\theta(t) = \frac{z_+(t)^{\odot 2} - z_-(t)^{\odot 2}}{4}$, implying that

$$
\begin{aligned}
\theta(t) &= \frac{z_+(0)^{\odot 2} e^{2\xi(t)} - z_-(0)^{\odot 2} e^{-2\xi(t)}}{4} \\
&= \frac{|z_+(0) \odot z_-(0)|}{2} \sinh\left(2\xi(t) + \log\left|\frac{z_+(0)}{z_-(0)}\right|\right).
\end{aligned}
\tag{26}
$$

## A.3  STUDY OF A DEEP REDUNDANT LINEAR NETWORKS

In this section, we emphasize that the strategy stated in Section A.1 is general enough to be applied to other similar parameterizations. We briefly analyze the implicit features of a Deep Redundant Linear Network defined through similar computations. The stated results can be seen as a simplified version of that obtained in Woodworth et al. (2020); Moroshko et al. (2020); Chou et al. (2023a) in the $L$-layer case.

We consider the parameterization $\theta = u^{\odot L}$ where the number of layers $L \in \mathbb{N}$ is strictly greater than 2. We also suppose that the loss function $\mathcal{L}$ can be written as $\mathcal{L}(\theta) = \ell(X\theta - y)$ for any $\theta \in \mathbb{R}^d$, with $\ell : \mathbb{R}^n \to \mathbb{R}$ which reaches its minimum at 0.

By applying a Gradient Flow to $\mathcal{L}(\theta)$ according to the vector of hyperparameters $u$, it holds that

$$\dot{u}(t) + Lu(t)^{\odot(L-1)} \odot \nabla\mathcal{L}(\theta(t)) = 0. \tag{27}$$

Supposing that for any $i \in \mathbb{R}^d$, $u_i(t)$ is different from 0 for any $t \geqslant 0$ (which can be done by enforcing positivity of $X$, $y$ and the initialization), we can write that

$$u(t) = \left(u(0)^{\odot-(L-2)} - L(L-2)\xi(t)\right)^{\odot-\frac{1}{L-2}}, \tag{28}$$

where $\xi : t \mapsto -\int_0^t \nabla\mathcal{L}(\theta(s))ds$. The original variable $\theta$ then satisfies:

$$\theta(t) = \left(u(0)^{\odot-(L-2)} - L(L-2)\xi(t)\right)^{\odot-\frac{L}{L-2}} =: \Psi\left(L(L-2)\,\xi(t)\right). \tag{29}$$

The inverse of $\Psi$ is trivial to compute as for any $\theta$, $\Psi^{-1}(\theta) = u(0)^{\odot-(L-2)} - \theta^{\odot-\left(1-\frac{2}{L}\right)}$. Then, simple computations show that the entropy function $\mathcal{Q}$ defined as:

$$\mathcal{Q}(\theta) = \langle u(0)^{\odot(L-2)}, \theta\rangle - \frac{L}{2}\left\langle \mathbf{1}, \theta^{\odot\frac{2}{L}}\right\rangle, \tag{30}$$

satisfies $\nabla\mathcal{Q}(\theta) = \Psi^{-1}(\theta)$ for any $\theta$. Then, by applying the same arguments as in the previous section, we get that:

$$\frac{d\nabla\mathcal{Q}(\theta(t))}{dt} + \nabla\mathcal{L}(\theta(t)) = 0, \tag{31}$$

and, if the trajectory converges to $\theta_\infty$ a solution of the problem, i.e. $X\theta_\infty = y$, then

$$\theta_\infty = \arg\min_{X\theta=y} \mathcal{Q}(\theta). \tag{32}$$

## A.4  DETAILED PROOF OF THEOREM 1

The proof of Theorem 1 follows the steps described in Section 2.2.
· We start by defining $\mathcal{M}$ in the following way:

$$\mathcal{M} = \left\{w \in \mathbb{R}^{L\times d}, \forall i \in [d], \#\left\{j \in [L], w_{(i-1)L+(j-1)} = 0\right\} \leqslant 1\right\}. \tag{33}$$

We can easily show that this set is a smooth submanifold of $\mathbb{R}^{L\times d}$. Let us rewrite $\mathcal{M}$ in the following way:

$$\mathcal{M} = \bigcap_{i\in[d]} \left\{w \in \mathbb{R}^{L\times d}, \#\left\{j \in [L], w_{(i-1)L+(j-1)} = 0\right\} \leqslant 1\right\}.$$

As a consequence, the complement set of $\mathcal{M}$ in $\mathbb{R}^{L \times d}$ denoted $\mathcal{M}^C$ satisfies:

$$\mathcal{M}^C = \bigcup_{i \in [d]} \left\{ w \in \mathbb{R}^{L \times d}, \ \exists (j,k) \in [L]^2, \ j \neq k, \ w_{(i-1)L+(j-1)} = w_{(i-1)L+(k-1)} = 0 \right\}$$

$$= \bigcup_{i \in [d]} \bigcup_{(j,k) \in [L]^2, \ j \neq k} \left\{ w \in \mathbb{R}^{L \times d}, \ w_{(i-1)L+(j-1)} = w_{(i-1)L+(k-1)} = 0 \right\}.$$

We can deduce that $\mathcal{M}^C$ is a closed set and that $\mathcal{M}$ is a smooth submanifold of $\mathbb{R}^{L \times d}$.

One can notice that if $\theta$ is parameterized by $\left(u^j\right)_{j \in [L]}$ satisfying $\mathcal{A}$, then the corresponding $\boldsymbol{u}(0)$ belongs to $\mathcal{M}$.

· We now prove that $G : \mathcal{M} \to \mathbb{R}^d$ is a commuting parameterization. Recall that it is said to be commuting on $\mathcal{M}$ if for any $(i_1, i_2) \in [d] \times [d]$ and $w \in \mathcal{M}$,

$$\nabla^2 G_{i_1}(w) \nabla G_{i_2}(w) - \nabla^2 G_{i_2}(w) \nabla G_{i_1}(w) = 0.$$

We have that for any $w \in \mathcal{M}$ and $i \in [d]$,

$$\forall j \in [Ld], \ \left(\nabla G_i(w)\right)_j = \begin{cases} \displaystyle\prod_{k=(i-1)L+1, \ k \neq j}^{iL} w_k & \text{if } j \in [\![(i-1)L+1, iL]\!], \\ 0 & \text{else.} \end{cases} \tag{34}$$

Thus, for any $i \in [d]$, $\left(\nabla^2 G_i(w)\right)_{jk} = 0$ as long as $j \notin [\![(i-1)L+1, iL]\!]$ or $k \notin [\![(i-1)L+1, iL]\!]$. As a consequence, for any $(i_1, i_2) \in [d] \times [d]$ such that $i_1 \neq i_2$, $\nabla^2 G_{i_1}(w) \nabla G_{i_2}(w) = 0$. We can deduce that $G$ is a commuting parameterization on $\mathcal{M}$.

· It is also required that $G$ is a regular parameterization on $\mathcal{M}$. More specifically, we need to show that $J_G(w)$ is of rank $d$ for all $w \in \mathcal{M}$.

We can easily obtain $J_G(w)$ from equation 34 and one can observe that it has the following structure:

$$J_G(w) = \begin{bmatrix} K_1^1 & \cdots & K_L^1 & 0 & \cdots & 0 & \cdots & 0 & \cdots & 0 \\ 0 & \cdots & 0 & K_1^2 & \cdots & K_L^2 & \cdots & 0 & \cdots & 0 \\ \vdots & \ddots & \vdots & \vdots & \ddots & \vdots & \ddots & \vdots & \ddots & \vdots \\ 0 & \cdots & 0 & 0 & \cdots & 0 & \cdots & K_1^d & \cdots & K_L^d \end{bmatrix},$$

where for any $(i, j) \in [d] \times [L]$,

$$K_j^i = \prod_{k=(i-1)L+1, \ k \neq j+L(i-1)}^{iL} w_k.$$

Hence, if for any $i \in [d]$ there exists $j \in [L]$ such that $K_j^i \neq 0$, then $J_G(w)$ is of rank $d$ and $G$ is a regular parameterization. Following the definition of $\mathcal{M}$, this condition is satisfied for all $w \in \mathcal{M}$ which ensures that $G$ is regular on $\mathcal{M}$.

· We finally need to check that (Li et al., 2022, Assumption 3.5) is satisfied by $G$, i.e that the domain of the flows induced by its gradient vector fields is pairwise symmetric. We refer the curious reader to the reference for more details, but we simply notice that since $\nabla G_j(w)$ and $\nabla G_k(w)$ have non zero entries on disjoint sets of components (if $j \neq k$), this assumption is directly satisfied.

From this point, we apply (Li et al., 2022, Theorem 4.8) and conclude.

