# OpenReview forum: "Optimization Insights into Deep Diagonal Linear Networks"
_ICLR.cc/2025/Conference — ICLR 2025 Conference Withdrawn Submission_

### Official Review · Reviewer_N7yG · 2024-10-22

**Soundness:** 2
**Presentation:** 3
**Contribution:** 1
**Rating:** 3
**Confidence:** 5

**Summary:**

This paper attempts to build the implicit regularization of gradient flow (GF) for the deep diagonal linear networks. In particular, the authors show that the GF dynamics induces a certain kind of mirror flow dynamics without an explicit form of the corresponding entropy function. In addition, they also reveal the convergence property of the GF dynamics and establish how the convergence rate is affected by the initialization.

**Strengths:**

In general, this paper is well organized, e.g., the authors clearly demonstrate their motivation and contribution. They also clearly develop their notations, definitions, and theorems to support their claims. These efforts make the understanding of this paper fairly straightforward. In addition,  the characterization of certain properties of the learning dynamics of deep diagonal linear networks might also be interesting, e.g., the second point of Proposition 1.

**Weaknesses:**

Unfortunately, both the technical and theoretical contributions of this paper are rather limited, which I will discuss as follows.

1. The ultimate goal of this paper is to reveal the implicit regularization effect of GF for deep diagonal linear networks. However, the explicit form of the corresponding entropy function for the induced mirror flow dynamics is completely absent. There is even no suggestion about possible properties that the entropy function should have.

    In addition, the derivation of the mirror flow form is a direct application of results in Li et al., 2022, and the first point of Proposition 1 can be a direct application of the Euler’s theorem for homogeneous function. Thus I think the technical contributions of this paper are rather limited.
2. As a comparison, Yun et al., 2021 already explicitly characterized the implicit bias of deep diagonal linear networks by using the tensor network formulation developed in their paper.

   Specifically, they established the optimization problem with an explicit form of the entropy function that the GF dynamics of deep diagonal linear networks (note that they did not require the parameterization $u^{\odot L} - v^{\odot L}$) aims to solve. They also established the convergence of the dynamics. The only possible weakness of their result is the additional requirement of the initialization, which the authors in this paper are able to relax at the cost of the characterization for explicit form of entropy function. But I cannot view such relaxation as a significant theoretical contribution that is sufficient for this paper to be published in its current version.

**Reference**

Yun et al., 2022. A Unifying View on Implicit Bias in Training Linear Neural Networks.

**Questions:**

1. What are the technical and theoretical contributions of this paper compared to Yun et al., 2021? For example, are results in this paper more general than those in Yun et al., 2021?
2. Can you derive the explicit form of the entropy function of the induced mirror flow dynamics?

---

### Official Review · Reviewer_WDCE · 2024-10-31

**Soundness:** 2
**Presentation:** 2
**Contribution:** 2
**Rating:** 3
**Confidence:** 2

**Summary:**

This work examines deep diagonal linear networks and demonstrates that gradient flow induces a mirror flow dynamic within the model. Under a mild initialization assumption, applying gradient flow to the layers generates a parameter trajectory that satisfies a mirror flow. The convergence properties related to the original optimization problem are highlighted.

**Strengths:**

The study brings a relatively fresh perspective of implicit bias.

**Weaknesses:**

-

**Questions:**

- For experiments, have you tested networks with different numbers of layers?\
- Line 480: "The initial value of the remaining layers of the first network (in blue) is generated randomly". Any requirements for the random initialization?

---

### Official Review · Reviewer_unKD · 2024-10-31

**Soundness:** 3
**Presentation:** 2
**Contribution:** 1
**Rating:** 3
**Confidence:** 5

**Summary:**

This paper shows gradient flow on diagonal linear networks induces a mirror flow on the input-output model, and also show that gradient flow converges exponentially given certain initializations.

**Strengths:**

The presentation in this paper is good, derivations are clear, and theoretical results are well explained.

**Weaknesses:**

This paper lacks novelty in the following ways:

1. Theorem 1, showing that GF on $L$-layer diagonal linear networks induces a mirror flow, as authors have acknowledged, is an application of Li et. al., 2022. So the contribution of this theorem is rather weak.

2. Theorem 2 is not new. Min et. al., 2023 (See their Section 4.2) have shown the exponential convergence of GF under the same condition as described in equation $(\mathcal{A})$ with a better lower bound on the rate.

**References**:

Z Li, et. al., Implicit Bias of Gradient Descent on Reparametrized Models: On Equivalence to Mirror Descent, NeurIPS, 2022

H Min, et. al., On the Convergence of Gradient Flow on Multi-layer Linear Models, ICML, 2023

**Questions:**

See "Weaknesses"

---

### Official Review · Reviewer_bXWM · 2024-11-04

**Soundness:** 3
**Presentation:** 2
**Contribution:** 2
**Rating:** 3
**Confidence:** 4

**Summary:**

The paper investigates the optimization dynamics of deep diagonal linear networks. Diagonal linear networks have garnered significant theoretical interest because they are easier to analyze and reflect many empirical behaviors induced by various hyper-parameters while training deep networks. The paper aims to diagonal networks to understand the impact of depth.

**Strengths:**

- The paper introduces a mild technical assumption $\mathcal{A}$ on the initialization which holds almost surely for a random initialization. Under this assumption, the gradient flow for the parameterization of deep diagonal network can be rewritten as mirror flow with a *convex* potential on the linear predictor $\theta$, which is an interesting technical observation.
- Under the same assumption, the linear convergence for any loss function $L$ which satisfies $PL$ condition in $\theta$ is established.

**Weaknesses:**

- The major weakness is the mirror potential is not explicitly defined - even if it not explicitly defined the limiting behavior in the case of large depth or small initialization are not discussed or analyzed which is a major drawback.
- The implicit bias of optimization benefits/drawbacks of the depth is not discussed and this weakens the motivation for studying the deep diagonal linear networks.

**Questions:**

1. Can the authors compare with convergence analysis and result for general deep linear networks in 1 ? Given the analysis of convergence for deep linear networks for general initializations, the convergence established for diagonal networks is insignificant.
2. Does the condition $\mathcal{A}$ *necessary* for the existence of convex potential ?


[1] Learning deep linear neural networks: Riemannian gradient flows and convergence to global minimizers Bah et. el. 2020

---

### Note · Authors · 2024-11-28

**Comment:**

We thank the reviewers for their valuable comments and choose to take the time to revise the manuscript before a possible resubmission.

**Withdrawal Confirmation:**

I have read and agree with the venue's withdrawal policy on behalf of myself and my co-authors.